# WHY GRADIENT CLIPPING ACCELERATES TRAINING: A THEORETICAL JUSTIFICATION FOR ADAPTIVITY

**Jingzhao Zhang, Tianxing He, Suvrit Sra & Ali Jadbabaie**
Massachusetts Institute of Technology
Cambridge, MA 02139, USA
{jzhzhang, tianxing, suvrit, jadbabai}@mit.edu

## ABSTRACT

We provide a theoretical explanation for the effectiveness of gradient clipping in training deep neural networks. The key ingredient is a new smoothness condition derived from practical neural network training examples. We observe that gradient smoothness, a concept central to the analysis of first-order optimization algorithms that is often assumed to be a constant, demonstrates significant variability along the training trajectory of deep neural networks. Further, this smoothness positively correlates with the gradient norm, and contrary to standard assumptions in the literature, it can grow with the norm of the gradient. These empirical observations limit the applicability of existing theoretical analyses of algorithms that rely on a fixed bound on smoothness. These observations motivate us to introduce a novel relaxation of gradient smoothness that is weaker than the commonly used Lipschitz smoothness assumption. Under the new condition, we prove that two popular methods, namely, *gradient clipping* and *normalized gradient*, converge arbitrarily faster than gradient descent with fixed stepsize. We further explain why such adaptively scaled gradient methods can accelerate empirical convergence and verify our results empirically in popular neural network training settings.

## 1 INTRODUCTION

We study optimization algorithms for neural network training and aim to resolve the mystery of why adaptive methods converge fast. Specifically, we study gradient-based methods for minimizing a differentiable nonconvex function $f : \mathbb{R}^d \to \mathbb{R}$, where $f(x)$ can potentially be stochastic, i.e., $f(x) = \mathbb{E}_\xi[F(x, \xi)]$. Such choices of $f$ cover a wide range of problems in machine learning, and their study motivates a vast body of current optimization literature.

A widely used (and canonical) approach for minimizing $f$ is the (stochastic) gradient descent (GD) algorithm. Despite its simple form, GD often achieves superior empirical (Wilson et al., 2017) performances and theoretical (Carmon et al., 2017) guarantees. However, in many tasks such as reinforcement learning and natural language processing (NLP), adaptive gradient methods (e.g., Adagrad (Duchi et al., 2011), ADAM (Kingma and Ba, 2014), and RMSProp (Tieleman and Hinton, 2012)) outperform SGD. Despite their superior empirical performance, our understanding of the fast convergence of adaptive methods is limited. Previous analysis has shown that adaptive methods are more robust to variation in hyper-parameters (Ward et al., 2018) and adapt to sparse gradients (Duchi et al., 2011) (a more detailed literature review is in Appendix A). However, in practice, the gradient updates are dense, and even after extensively tuning the SGD hyperparameters, it still converges much slower than adaptive methods in NLP tasks.

We analyze the convergence of clipped gradient descent and provide an explanation for its fast convergence. Even though gradient clipping is a standard practice in tasks such as language models (e.g. Merity et al., 2018; Gehring et al., 2017; Peters et al., 2018), it lacks a firm theoretical grounding. Goodfellow et al. (2016); Pascanu et al. (2013; 2012) discuss the gradient explosion problem in recurrent models and consider clipping as an intuitive work around. We formalize this intuition and prove that clipped GD can *converge arbitrarily faster than fixed-step gradient descent*. This result is shown to hold under a novel smoothness condition that is *strictly weaker* than the standard Lipschitz-gradient assumption pervasive in the literature. Hence our analysis captures many functions that are not globally Lipschitz smooth. Importantly, the proposed smoothness condition is derived on the

basis of extensive NLP training experiments, which are precisely the same type of experiments for which adaptive gradient methods empirically perform superior to gradient methods.

By identifying a a new smoothness condition through experiments and then using it to analyze the convergence of adaptively-scaled methods, we reduce the following gap between theory and practice. On one hand, powerful techniques such as Nesterov's momentum and variance reduction theoretically accelerate convex and nonconvex optimization. But, at least for now, they seem to have limited applicability in deep learning (Defazio and Bottou, 2018). On the other hand, some widely used techniques (e.g., heavy-ball momentum, adaptivity) lack theoretical acceleration guarantees. We suspect that a major reason here is the misalignment of the theoretical assumptions with practice. Our work demonstrates that the concept of acceleration critically relies on the problem assumptions and that the standard global Lipschitz-gradient condition may not hold in the case of some applications and thus must be relaxed to admit a wider class of objective functions.

## 1.1 CONTRIBUTIONS

In light of the above background, the main contributions of this paper are the following:

- Inspired and supported by neural network training experiments, we introduce a new smoothness condition that allows the local smoothness constant to increase with the gradient norm. This condition is *strictly weaker* than the pervasive Lipschitz-gradient assumption.
- We provide a convergence rate for clipped GD under our smoothness assumption (Theorem 3).
- We prove an upper-bound (Theorem 6) and a lower-bound (Theorem 4) on the convergence rate of GD under our relaxed smoothness assumption. The lower-bound demonstrates that GD with fixed step size can be *arbitrarily slower* than clipped GD.
- We provide upper bounds for stochastic clipped GD (Theorem 7) and SGD (Theorem 8). Again, stochastic clipped GD can be arbitrarily faster than SGD with a fixed step size.

We support our proposed theory with realistic neural network experiments. First, in the state of art LSTM language modeling (LM) setting, we observe the function smoothness has a strong correlation with gradient norm (see Figure 2). This aligns with the known fact that gradient clipping accelerates LM more effectively compared to computer vision (CV) tasks. Second, our experiments in CV and LM demonstrate that clipping accelerates training error convergence and allows the training trajectory to cross non-smooth regions of the loss landscape. Furthermore, gradient clipping can also achieve good generalization performance even in image classification (e.g., $95.2\%$ test accuracy in 200 epochs for ResNet20 on Cifar10). Please see Section 5 for more details.

## 2 A NEW RELAXED SMOOTHNESS CONDITION

In this section, we motivate and develop a relaxed smoothness condition that is weaker (and thus, more general) than the usual global Lipschitz smoothness assumption. We start with the traditional definition of smoothness.

### 2.1 FUNCTION SMOOTHNESS (LIPSCHITZ GRADIENTS)

Recall that $f$ denotes the objective function that we want to minimize. We say that $f$ is $L$-smooth if

$$\|\nabla f(x) - \nabla f(y)\| \leq L\|x - y\|, \quad \text{for all } x, y \in \mathbb{R}^d. \tag{1}$$

For twice differentiable functions, condition (1) is equivalent to $\|\nabla^2 f(x)\| \leq L, \forall x \in \mathbb{R}^d$. This smoothness condition enables many important theoretical results. For example, Carmon et al. (2017) show that GD with $h = 1/L$ is up to a constant optimal for optimizing smooth nonconvex functions.

But the usual $L$-smoothness assumption (1) also has its limitations. Assuming existence of a global constant $L$ that upper bounds the variation of the gradient is very restrictive. For example, simple polynomials such as $f(x) = x^3$ break the assumption. One workaround is to assume that $L$ exists in a compact region, and either prove that the iterates do not escape the region or run projection-based algorithms. However, such assumptions can make $L$ very large and slow down the theoretical convergence rate. In Section 4, we will show that a slow rate is unavoidable for gradient descent with fixed step size, whereas clipped gradient descent can greatly improve the dependency on $L$.

The above limitations force fixed-step gradient descent (which is tailored for Lipschitz smooth functions) to converge slowly in many tasks. In Figure 1, we plot the estimated function smoothness at different iterations during training neural networks. We find that function smoothness varies greatly at different iterations. From Figure 1, we further find that local smoothness positively correlates with the full gradient norm, especially in the language modeling experiment. A natural question is:

> *Can we find a fine-grained smoothness condition under which we can design theoretically and empirically fast algorithms at the same time?*

To answer this question, we introduce the relaxed smoothness condition in the next section, which is developed on the basis of extensive experiments— Figure 1 provides an illustrative example.

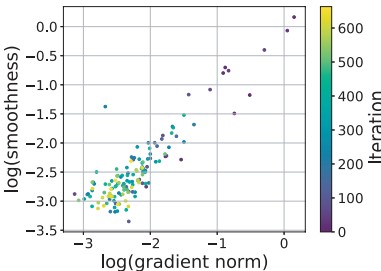

Figure 1: Gradient norm vs local gradient Lipschitz constant on a log-scale along the training trajectory for AWD-LSTM (Merity et al., 2018) on PTB dataset. The colorbar indicates the number of iterations during training. More experiments can be found in Section 5. Experiment details are in Appendix H.

## 2.2 A NEW RELAXED SMOOTHNESS CONDITION

We observe strong positive correlation between function smoothness and gradient norm in language modeling experiments (Figure 1(a)). This observation leads us to propose the following smoothness condition that allows local smoothness to grow with function gradients.

**Definition 1.** A second order differentiable function $f$ is $(L_0, L_1)$-smooth if

$$\|\nabla^2 f(x)\| \leq L_0 + L_1 \|\nabla f(x)\|. \tag{2}$$

Definition 1 *strictly relaxes* the usual (and widely used) $L$-smoothness. There are two ways to interpret the relaxation: First, when we focus on a compact region, we can balance the constants $L_0$ and $L_1$ such that $L_0 \ll L$ while $L_1 \ll L$. Second, there exist functions that are $(L_0, L_1)$-smooth globally, but not $L$-smooth. Hence the constant $L$ for $L$-smoothness gets larger as the compact set increases but $L_0$ and $L_1$ stay fixed. An example is given in Lemma 2.

**Remark 1.** It is worth noting that we do not need the Hessian operator norm and gradient norm to necessarily satisfy the linear relation (2). As long as these norms are positively correlated, gradient clipping can be shown to achieve faster rate than fixed step size gradient descent. We use the linear relationship (2) for simplicity of exposition.

**Lemma 2.** *Let $f$ be the univariate polynomial $f(x) = \sum_{i=1}^{d} a_i x^i$. When $d \geq 3$, then $f$ is $(L_0, L_1)$-smooth for some $L_0$ and $L_1$ but not $L$-smooth.*

*Proof.* The first claim follows from $\lim_{x \to \infty} \left| \frac{f'(x)}{f''(x)} \right| = \lim_{x \to -\infty} \left| \frac{f'(x)}{f''(x)} \right| = \infty$. The second claim follows by the unboundedness of $f''(x)$. □

## 2.3 SMOOTHNESS IN NEURAL NETWORKS

We saw that our smoothness condition relaxes the traditional smoothness assumption and is motivated empirically (Figure 1). Below we develop some intuition for this phenomenon. We conjecture that the proposed positive correlation results from the common components in expressions of the gradient and the Hessian. We illustrate the reasoning behind this conjecture by considering an $\ell$-layer linear network with quadratic loss—a similar computation also holds for nonlinear networks.

The $L_2$ regression loss of a deep linear network is $\mathcal{L}(Y, f(X)) := \|Y - W_\ell \cdots W_1 X\|^2$, where $Y$ denotes labels, $X$ denotes the input data matrix, and $W_i$ denotes the weights in the $i^{\text{th}}$ layer. By (Lemma 4.3 Kawaguchi, 2016), we know that

$$\nabla_{\text{vec}(w_i)} \mathcal{L}(Y, f(X)) = ((W_\ell \cdots W_{i+1}) \otimes (W_{i-1} \cdots W_2 W_1 X)^T)^T \text{vec}(f(X) - Y),$$

where $\text{vec}(\cdot)$ flattens a matrix in $\mathbb{R}^{m \times n}$ into a vector in $\mathbb{R}^{mn}$; $\otimes$ denotes the Kronecker product. For constants $i, j$ such that $\ell \geq j > i > 0$, the second order derivative

$$\nabla_{\text{vec}(w_j)} \nabla_{\text{vec}(w_i)} \mathcal{L}(Y, f(X)) =$$
$$((W_\ell \cdots W_{i+1}) \otimes (W_{i-1} \cdots W_2 W_1 X)^T)^T ((W_\ell \cdots W_{j+1}) \otimes (W_{j-1} \cdots W_2 W_1 X)^T) +$$
$$((W_{j-1} \cdots W_{i+1}) \otimes (W_{i-1} \cdots W_2 W_1 X))(I \otimes ((f(X) - Y)W_\ell \cdots W_{j+1})).$$

When $j = i$, the second term equals 0. Based on the above expressions, we notice that the gradient norm and Hessian norm may be positively correlated due to the following two observations. First, the gradient and the Hessian share many components such as the matrix product of weights across layers. Second, if one naively upper bounds the norm using Cauchy-Schwarz, then both upper-bounds would be monotonically increasing with respect to $\|W_i\|$ and $\|f(X) - Y\|$.

## 3 PROBLEMS SETUP AND ALGORITHMS

In this section, we state the optimization problems and introduce gradient based algorithms for them that work under the new smoothness condition (2). Convergence analysis follows in Section 4.

Recall that we wish to solve the nonconvex optimization problem $\min_{x \in \mathbb{R}^d} f(x)$. Since in general this problem is intractable, following common practice we also seek an $\epsilon$-stationary point, i.e., a point $x$ such that $\|\nabla f(x)\| \leq \epsilon$. Furthermore, we make the following assumptions to regularize the function class studied and subsequently provide nonasymptotic convergence rate analysis.

**Assumption 1.** The function $f$ is lower bounded by $f^* > -\infty$.

**Assumption 2.** The function $f$ is twice differentiable.

**Assumption 3** $((L_0, L_1)$**-smoothness).** The function $f$ is $(L_0, L_1)$-smooth, i.e., there exist positive constants $L_0$ and $L_1$ such that $\|\nabla^2 f(x)\| \leq L_0 + L_1 \|\nabla f(x)\|$—see condition (2).

The first assumption is standard. Twice differentiability in Assumption 2 can relaxed to first-order differentiability by modifying the definition of $(L_0, L_1)$-smoothness as

$$\limsup_{\delta \to \vec{0}} \frac{\|\nabla f(x) - \nabla f(x+\delta)\|}{\|\delta\|} \leq L_1 \|\nabla f(x)\| + L_0.$$

The above inequality implies $\nabla f(x)$ is locally Lipschitz, and hence almost everywhere differentiable. Therefore, all our results can go through by handling the integrations more carefully. But to avoid complications and simplify exposition, we assume that the function is twice differentiable.

To further relax the global assumptions, by showing that GD and clipped GD are monotonically decreasing in function value, we require the above assumptions to hold just in a neighborhood determined by the sublevel set $\mathcal{S}^1$ for a given initialization $x_0$, where

$$\mathcal{S} := \{x \mid \exists \, y \text{ such that } f(y) \leq f(x_0), \text{ and } \|x - y\| \leq 1\}. \tag{3}$$

### 3.1 GRADIENT DESCENT ALGORITHMS

In this section, we review a few well-known variants of gradient based algorithms that we analyze. We start with the ordinary *gradient descent* with a fixed step size $\eta$,

$$x_{k+1} = x_k - \eta \nabla f(x_k). \tag{4}$$

This algorithm (pedantically, its stochastic version) is widely used in neural network training. Many modifications of it have been proposed to stabilize or accelerate training. One such technique of particular importance is *clipped gradient descent*, which performs the following updates:

$$x_{k+1} = x_k - h_c \nabla f(x_k), \quad \text{where } h_c := \min\{\eta_c, \frac{\gamma \eta_c}{\|\nabla f(x)\|}\}. \tag{5}$$

---

[1]The constant "1" in the expression (3) is arbitrary and can be replaced by any fixed positive constant.

Another algorithm that is less common in practice but has attracted theoretical interest is *normalized gradient descent*. The updates for normalized GD method can be written as

$$x_{k+1} = x_k - h_n \nabla f(x_k), \quad \text{where } h_n := \frac{\eta_n}{\|\nabla f(x)\| + \beta}.$$ (6)

The stochastic version of the above algorithms replace the gradient with a stochastic estimator.

We note that Clipped GD and NGD are almost equivalent. Indeed, for any given $\eta_n$ and $\beta$, if we set $\gamma \eta_c = \eta_n$ and $\eta_c = \eta_n / \beta$, then we have

$$\tfrac{1}{2} h_c \leq h_n \leq 2 h_c.$$

Therefore, clipped GD is equivalent to NGD up to a constant factor in the step size choice. Consequently, the nonconvex convergence rates in Section 4 and Section 4.2 for clipped GD also apply to NGD. We omit repeating the theorem statements and the analysis for conciseness.

## 4 THEORETICAL ANALYSIS

In this section, we analyze the oracle complexities of GD and clipped GD under our relaxed smoothness condition. All the proofs are in the appendix. We highlight the key theoretical challenges that needed to overcome in Appendix B (e.g., due to absence of Lipschitz-smoothness, already the first-step of analysis, the so-called "descent lemma" fails).

Since we are analyzing the global iteration complexity, let us recall the formal definition being used. We follow the notation from Carmon et al. (2017). For a deterministic sequence $\{x_k\}_{k \in \mathbb{N}}$, define the complexity of $\{x_k\}_{k \in \mathbb{N}}$ for a function $f$ as

$$T_\epsilon(\{x_t\}_{t \in \mathbb{N}}, f) := \inf\{t \in \mathbb{N} | \|\nabla f(x_t)\| \leq \epsilon\}.$$ (7)

For a random process $\{x_k\}_{k \in \mathbb{N}}$, we define the complexity of $\{x_k\}_{k \in \mathbb{N}}$ for function $f$ as

$$T_\epsilon(\{x_t\}_{t \in \mathbb{N}}, f) := \inf\Big\{t \in \mathbb{N} | \text{Prob}(\|\nabla f(x_k)\| \geq \epsilon \text{ for all } k \leq t) \leq \tfrac{1}{2}\Big\}.$$ (8)

In particular, if the condition is never satisfied, then the complexity is $\infty$. Given an algorithm $A_\theta$, where $\theta$ denotes hyperparameters such as step size and momentum coefficient, we denote $A_\theta[f, x_0]$ as the sequence of (potentially stochastic) iterates generated by $A$ when operating on $f$ with initialization $x_0$. Finally, we define the iteration complexity of an algorithm class parameterized by $p$ hyperparameters, $\mathcal{A} = \{A_\theta\}_{\theta \in \mathbb{R}^p}$ on a function class $\mathcal{F}$ as

$$\mathcal{N}(\mathcal{A}, \mathcal{F}, \epsilon) := \inf_{A_\theta \in \mathcal{A}} \sup_{x_0 \in \mathbb{R}^d, f \in \mathcal{F}} T_\epsilon(A_\theta[f, x_0], f).$$ (9)

The definition in the stochastic setting simply replaces the expression (7) with the expression (8). In the rest of the paper, "iteration complexity" refers to the quantity defined above.

### 4.1 CONVERGENCE IN THE DETERMINISTIC SETTING

In this section, we present the convergence rates for GD and clipped GD under deterministic setting. We start by analyzing the **clipped GD** algorithm with update defined in equation (5).

**Theorem 3.** *Let $\mathcal{F}$ denote the class of functions that satisfy Assumptions 1, 2, and 3 in set $\mathcal{S}$ defined in (3). Recall $f^*$ is a global lower bound for function value. With $\eta_c = \frac{1}{10 L_0}, \gamma = \min\{\frac{1}{\eta_c}, \frac{1}{10 L_1 \eta_c}\}$, we can prove that the iteration complexity of clipped GD (Algorithm 5) is upper bounded by*

$$\frac{20 L_0 (f(x_0) - f^*)}{\epsilon^2} + \frac{20 \max\{1, L_1^2\}(f(x_0) - f^*)}{L_0}.$$

The proof of Theorem 3 is included in Appendix C.

Now, we discuss the convergence of vanilla GD. The standard GD is known to converge to first order $\epsilon$-stationary points in $\mathcal{O}((L(f(x_0) - f^*))\epsilon^{-2})$ iterations for $(L, 0)-$smooth nonconvex functions. By Theorem 1 of Carmon et al. (2017), this rate is up to a constant optimal.

However, we will show below that gradient descent is suboptimal under our relaxed $(L_0, L_1)$-smoothness condition. In particular, to prove the convergence rate for gradient descent with fixed step size, we need to permit it benefit from an additional assumption on gradient norms.

**Assumption 4.** Given an initialization $x_0$, we assume that
$$M := \sup\{\|\nabla f(x)\| \mid x \text{ such that } f(x) \leq f(x_0)\} < \infty.$$

This assumption is in fact *necessary*, as our next theorem reveals.

**Theorem 4.** *Let $\mathcal{F}$ be the class of objectives satisfying Assumptions 1, 2, 3, and 4 with fixed constants $L_0 \geq 1$, $L_1 \geq 1$, $M > 1$. The iteration complexity for the fixed-step gradient descent algorithms parameterized by step size $h$ is at least*
$$\frac{L_1 M(f(x_0) - f^* - 5\epsilon/8)}{8\epsilon^2(\log M + 1)}.$$

The proof can be found in Appendix D.

**Remark 5.** Theorem 1 of Carmon et al. (2017) and Theorem 4 together show that gradient descent with a fixed step size cannot converge to an $\epsilon$-stationary point faster than $\Omega\left((L_1 M/\log(M) + L_0)(f(x_0) - f^*)\epsilon^{-2}\right)$. Recall that clipped GD algorithm converges as $\mathcal{O}\left(L_0(f(x_0) - f^*)\epsilon^{-2} + L_1^2(f(x_0) - f^*)L_0^{-1}\right)$. Therefore, clipped GD can be arbitrarily faster than GD when $L_1 M$ is large, or in other words, when the problem has a *poor initialization*.

Below, we provide an iteration upper bound for the fixed-step gradient descent update (4).

**Theorem 6.** *Suppose assumptions 1, 2, 3 and 4 hold in set $\mathcal{S}$ defined in (3). If we pick parameters such that $h = \frac{1}{(2(ML_1 + L_0))}$, then we can prove that the iteration complexity of GD with a fixed step size defined in Algorithm 4 is upper bounded by*
$$4(ML_1 + L_0)(f(x_0) - f^*)\epsilon^{-2}.$$

Please refer to Appendix E for the proof. Theorem 6 shows that gradient descent with a fixed step size converges in $\mathcal{O}((ML_1 + L_0)(f(x_0) - f^*)/\epsilon^2)$ iterations. This suggests that the lower bound in Remark 5 is tight up to a log factor in $M$.

## 4.2 CONVERGENCE IN THE STOCHASTIC SETTING

In the stochastic setting, we assume GD and clipped GD have access to an unbiased stochastic gradient $\nabla \hat{f}(x)$ instead of the exact gradient $\nabla f(x)$. For simplicity, we denote $g_k = \nabla \hat{f}(x_k)$ below. To prove convergence, we need the following assumption.

**Assumption 5.** There exists $\tau > 0$, such that $\|\nabla \hat{f}(x) - \nabla f(x)\| \leq \tau$ almost surely.

Bounded noise can be relaxed to sub-gaussian noise if the noise is symmetric. Furthermore, up to our knowledge, this is the first stochastic nonconvex analysis of adaptive methods that does not require the gradient norm $\|\nabla f(x)\|$ to be bounded globally.

The main result of this section is the following convergence guarantee for stochastic clipped GD (based on the stochastic version of the update (5)).

**Theorem 7.** *Let Assumptions 1–3 and 5 hold globally with $L_1 > 0$. Let $h = \min\{\frac{1}{16\eta L_1(\|g_k\| + \tau)}, \eta\}$ where $\eta = \min\{\frac{1}{20L_0}, \frac{1}{128L_1\tau}, \frac{1}{\sqrt{T}}\}$. Then we can show that iteration complexity for stochastic clipped GD after of update (5) is upper bounded by*
$$\Delta \max\{\frac{128L_1}{\epsilon}, \frac{4\Delta}{\epsilon^4}, \frac{80L_0 + 512L_1\tau}{\epsilon^2}\},$$
*where $\Delta = (f(x_0) - f^* + (5L_0 + 2L_1\tau)\tau^2 + 9\tau L_0^2/L_1)$.*

In comparison, we have the following upper bound for ordinary SGD.

**Theorem 8.** *Let Assumptions 1–3, and 5 hold globally with $L_1 > 0$. Let $h = \min\{\frac{1}{\sqrt{T}}, \frac{1}{L_1(M+\tau)}\}$. Then the iteration complexity for the stochastic version of GD (4) is upper bounded by*
$$\left(f(x_0) - f^* + (5L_0 + 4L_1 M)(M + \tau)^2\right)^2 \epsilon^{-4}.$$

We cannot provide a lower bound for this algorithm. In fact, lower bound is not known for SGD even in the global smoothness setting. However, the deterministic lower bound in Theorem 4 is still valid, though probably loose. Therefore, the convergence of SGD still requires additional assumption and can again **be arbitrarily slower** compared to clipped SGD when $M$ is large.

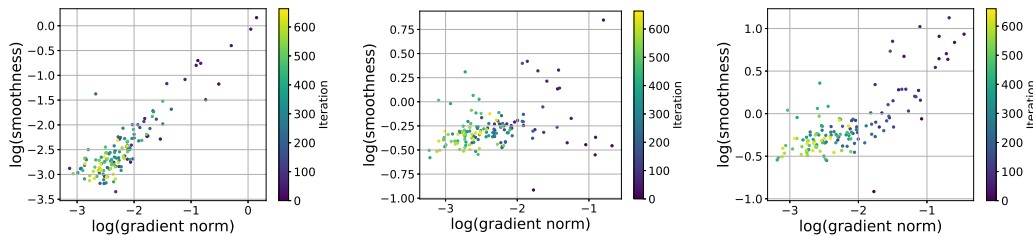

(a) Learning rate 30, with clipping.  (b) Learning rate 2, without clipping.  (c) Learning rate 2, with clipping.

Figure 2: Gradient norm vs smoothness on log scale for LM training. The dot color indicates the iteration number. Darker ones correspond to earlier iterations. Note that the spans of $x$ and $y$ axis are not fixed.

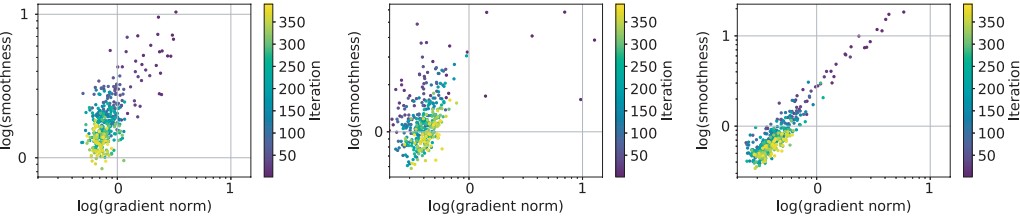

(a) SGD with momentum.    (b) Learning rate 1, without clipping.  (c) Learning rate 5, with clipping.

Figure 3: Gradient norm vs smoothness on log scale for ResNet20 training. The dot color indicates the iteration number.

## 5 EXPERIMENTS

In this section, we summarize our empirical findings on the positive correlation between gradient norm and local smoothness. We then show that clipping accelerates convergence during neural network training. Our experiments are based on two tasks: language modeling and image classification. We run language modeling on the Penn Treebank (PTB) (Mikolov et al., 2010) dataset with AWD-LSTM models (Merity et al., 2018)[2]. We train ResNet20 (He et al., 2016) on the Cifar10 dataset (Krizhevsky and Hinton, 2009). Details about the smoothness estimation and experimental setups are in Appendix H. An additional synthetic experiment is discussed in Appendix I.

First, our experiments test whether the local smoothness constant increases with the gradient norm, as suggested by the relaxed smoothness conditions defined in (2) (Section 2). To do so, we evaluate both quantities at points generated by the optimization procedure. We then scatter the local smoothness constants against the gradient norms in Figure 2 and Figure 3. Note that the plots are on a log-scale. A linear scale plot is shown in Appendix Figure 5.

We notice that the correlation exists in the default training procedure for language modeling (see Figure 2a) but not in the default training for image classification (see Figure 3a). This difference aligns with the fact that gradient clipping is widely used in language modeling but is less popular in ResNet training, *offering empirical support to our theoretical findings*.

We further investigate the cause of correlation. The plots in Figures 2 and 3 show that correlation appears when the models are trained with clipped GD and large learning rates. We propose the following explanation. Clipping enables the training trajectory to stably traverse non-smooth regions. Hence, we can observe that gradient norms and smoothness are positively correlated in Figures 2a and 3c. Without clipping, the optimizer has to adopt a small learning rate and stays in a region where local smoothness does not vary much, otherwise the sequence diverges, and a different learning rate is used. Therefore, in other plots of Figures 2 and 3, the correlation is much weaker.

As positive correlations are present in both language modeling and image classification experiments with large step sizes, our next set of experiments checks whether clipping helps accelerate convergence as predicted by our theory. From Figure 4, we find that clipping indeed accelerates convergence. Because gradient clipping is a standard practice in language modeling, the LSTM models trained with clipping achieve the best validation performance and the fastest training loss conver-

---

[2]Part of the code is available at https://github.com/JingzhaoZhang/why-clipping-accelerates

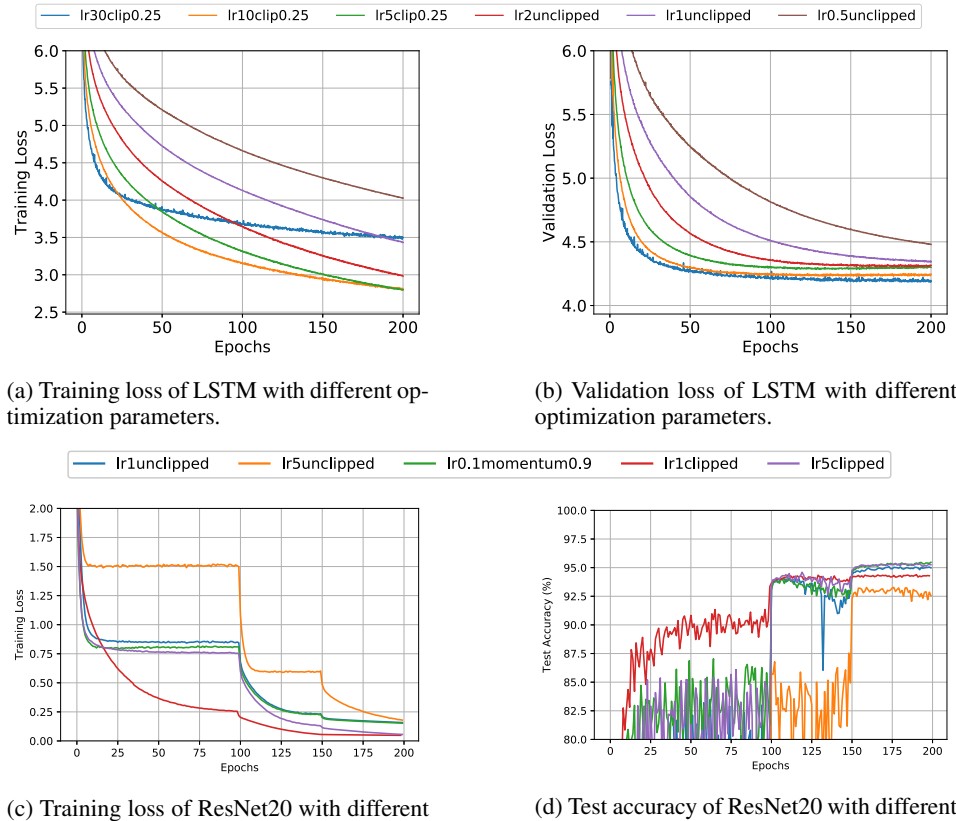

Figure 4: Training and validation loss obtained with different training methods for LSTM and ResNet training. The validation loss plots the cross entropy. The training loss additionally includes the weight regularization term. In the legend, 'lr30clip0.25' denotes that clipped SGD uses step size 30 and that the $L_2$ norm of the stochastic gradient is clipped by 0.25. In ResNet training, we threshold the stochastic gradient norm at 0.25 when clipping is applied.

gence as expected. For image classification, surprisingly, clipped GD also achieves the fastest convergence and matches the test performance of SGD+momentum. These plots show that clipping can accelerate convergence and achieve good test performance at the same time.

## 6  DISCUSSION

Much progress has been made to close the gap between upper and lower oracle complexities for first order smooth optimization. The works dedicated to this goal provide important insights and tools for us to understand the optimization procedures. However, there is another gap that separates theoretically *accelerated* algorithms from empirically fast algorithms.

Our work aims to close this gap. Specifically, we propose a relaxed smoothness assumption that is supported by empirical evidence. We analyze a simple but widely used optimization technique known as gradient clipping and provide theoretical guarantees that clipping can accelerate gradient descent. This phenomenon aligns remarkably well with empirical observations.

There is still much to be explored in this direction. First, though our smoothness condition relaxes the usual Lipschitz assumption, it is unclear if there is an even better condition that also matches the experimental observations while also enabling a clean theoretical analysis. Second, we only study convergence of clipped gradient descent. Studying the convergence properties of other techniques such as momentum, coordinate-wise learning rates (more generally, preconditioning), and variance reduction is also interesting. Finally, the most important question is: *"can we design fast algorithms based on relaxed conditions that achieve faster convergence in neural network training?"*

Our experiments also have noteworthy implications. First, though advocating clipped gradient descent in ResNet training is not a main point of this work, it is interesting to note that gradient

descent and clipped gradient descent with large step sizes can achieve a similar *test performance* as momentum-SGD. Second, we learned that the performance of the baseline algorithm can actually beat some recently proposed algorithms. Therefore, when we design or learn about new algorithms, we need to pay extra attention to check whether the baseline algorithms are properly tuned.

## 7 ACKNOWLEDGEMENT

SS acknowledges support from an NSF-CAREER Award (Number 1846088) and an Amazon Research Award. AJ acknowledges support from an MIT-IBM-Exploratory project on adaptive, robust, and collaborative optimization.

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

## A    MORE RELATED WORK ON ACCELERATING GRADIENT METHODS

**Variance reduction.**  Many efforts have been made to accelerate gradient-based methods.  One elegant approach is variance reduction (e.g. Schmidt et al., 2017; Johnson and Zhang, 2013; Defazio et al., 2014; Bach and Moulines, 2013; Konečnỳ and Richtárik, 2013; Xiao and Zhang, 2014; Gong and Ye, 2014; Fang et al., 2018; Zhou et al., 2018b).  This technique aims to solve stochastic and finite sum problems by averaging the noise in the stochastic oracle via utilizing the smoothness of the objectives.

**Momentum methods.**  Another line of work focuses on achieving acceleration with momentum. Polyak (1964) showed that momentum can accelerate optimization for quadratic problems; later, Nesterov (1983) designed a variation that provably accelerate any smooth convex problems. Based on Nesterov's work, much theoretical progress was made to accelerate different variations of the original smooth convex problems (e.g. Ghadimi and Lan, 2016; 2012; Beck and Teboulle, 2009; Shalev-Shwartz and Zhang, 2014; Jin et al., 2018; Carmon et al., 2018; Allen-Zhu, 2017; Lin et al., 2015; Nesterov, 2012).

**Adaptive step sizes.**  The idea of varying step size in each iteration has long been studied.  Armijo (1966) proposed the famous backtracking line search algorithm to choose step size dynamically. Polyak (1987) proposed a strategy to choose step size based on function suboptimality and gradient norm.  More recently, Duchi et al. (2011) designed the Adagrad algorithm that can utilize the sparsity in stochastic gradients.

Since 2018, there has been a surge in studying the theoretical properties of adaptive gradient methods. One starting point is (Reddi et al., 2019), which pointed out that ADAM is not convergent and proposed the AMSGrad algorithm to fix the problem.  Ward et al. (2018); Li and Orabona (2018) prove that Adagrad converges to stationary point for nonconvex stochastic problems. Zhou et al. (2018a) generalized the result to a class of algorithms named Padam. Zou et al. (2018); Staib et al. (2019); Chen et al. (2018); Zhou et al. (2018c); Agarwal et al. (2018); Zhou et al. (2018b); Zou and Shen (2018) also studied different interesting aspects of convergence of adaptive methods.  In addition, Levy (2016) showed that normalized gradient descent may have better convergence rate in presence of injected noise.  However, the rate comparison is under dimension dependent setting.  Hazan et al. (2015) studied the convergence of normalized gradient descent for quasi-convex functions.

## B    CHALLENGES IN THE PROOFS

In this section, we highlight a few key challenges in our proofs.  First, the analysis convergence under the relaxed smoothness condition is more difficult than the traditional setup.  In particular, classical analyses based on Lipschitz-smooth gradients frequently exploit the descent condition:

$$f(y) \le f(x) + \langle \nabla f(x), y - x \rangle + \tfrac{L}{2}\|y - x\|^2. \tag{10}$$

However, under our relaxed smoothness condition, the last term will increase exponentially in $\|y - x\|^2$. To solve this challenge, we bound the distance moved by clipping and apply Grönwall's inequality.

Second, our algorithm specific lower bound proved in Theorem 4 is novel and tight up to a log factor. To our knowledge, the worst case examples used have not been studied before.

Last, proving the convergence of adaptive methods in the nonconvex stochastic setting suffers from a fundamental challenge: the stochastic gradient is dependent on the update step size. This problem is usually circumvented by either assuming gradients have bounded norms or by using a lagging-by-one step-size to decouple the correlation.  The situation is even worse under the relaxed smoothness assumption.  In our case, we overcome this challenge by a novel analysis that divides the proof into the large gradient scenario and the small gradient scenario.

## C    PROOF OF THEOREM 3

We start by proving a lemma that is repeatedly used in later proofs. The lemma bounds the gradient in a neighborhood of the current point by Grönwall's inequality (integral form).

**Lemma 9.** *Given $x$ such that $f(x) \leq f(x_0)$, for any $x^+$ such that $\|x^+ - x\| \leq \min\{1/L_1, 1\}$, we have $\|\nabla f(x^+)\| \leq 4(L_0/L_1 + \|\nabla f(x)\|)$.*

**Remark 10.** Note that the constant "1" comes from the definition of $\mathcal{S}$ in (3). If Assumption 3 holds globally, then we do not need to constrain $\|x^+ - x\| \leq 1$. This version will be used in Theorem 7.

*Proof.* Let $\gamma(t)$ be a curve defined below,

$$\gamma(t) = t(x^+ - x) + x, \ t \in [0, 1].$$

Then we have

$$\nabla f(\gamma(t)) = \int_0^t \nabla^{(2)} f(\gamma(\tau))(x^+ - x)d\tau + \nabla f(\gamma(0)).$$

By Cauchy-Schwarz's inequality, we get

$$\|\nabla f(\gamma(t))\| \leq \|x^+ - x\| \int_0^t \|\nabla^{(2)} f(\gamma(\tau))\|d\tau + \|\nabla f(x)\|$$

$$\leq \frac{1}{L_1} \int_0^t (L_0 + L_1 \|\nabla f(\gamma(\tau))\|)d\tau + \|\nabla f(x)\|.$$

The second inequality follows by Assumption 3. Then we can apply the integral form of Grönwall's inequality and get

$$\|\nabla f(\gamma(t))\| \leq \frac{L_0}{L_1} + \|\nabla f(x)\| + \int_0^t \left(\frac{L_0}{L_1} + \|\nabla f(x)\|\right) \exp(t - \tau)d\tau.$$

The Lemma follows by setting $t = 1$. $\qquad\square$

## C.1 PROOF OF THE THEOREM

We parameterize the path between $x_k$ and its updated iterate $x_{k+1}$ as follows:

$$\gamma(t) = t(x_{k+1} - x_k) + x_k, \forall t \in [0, 1].$$

Since $x_{k+1} = x_k - h_k \nabla f(x_k)$, using Taylor's theorem, the triangle inequality, and Cauchy-Schwarz, we obtain

$$f(x_{k+1}) \leq f(x_k) - h_k \|\nabla f(x_k)\|^2 + \frac{\|x_{k+1} - x_k\|^2}{2} \int_0^1 \|\nabla^2 f(\gamma(t))\|dt.$$

Since

$$h_k \leq \frac{\gamma\eta}{\|\nabla f(x)\|} \leq \min\left\{\frac{1}{\|\nabla f(x)\|}, \frac{1}{L_1 \|\nabla f(x_k)\|}\right\},$$

we know by Lemma 9

$$\|\nabla f(\gamma(t))\| \leq 4(\tfrac{L_0}{L_1} + \|\nabla f(x)\|).$$

Then by Assumption 3, we obtain the "descent inequality":

$$f(x_{k+1}) \leq f(x_k) - h_k \|\nabla f(x_k)\|^2 + \frac{5L_0 + 4L_1 \|\nabla f(x_k)\|}{2} \|\nabla f(x_k)\|^2 h_k^2.$$

Therefore, as long as $h_k \leq 1/(5L_0 + 4L_1 \|\nabla f(x_k)\|)$ (which follows by our choice of $\eta, \gamma$), we can quantify the descent to be

$$f(x_{k+1}) \leq f(x_k) - \frac{h_k \|\nabla f(x_k)\|^2}{2}.$$

When $\|\nabla f(x_k)\| \geq L_0/L_1$, we have

$$\frac{h_k \|\nabla f(x_k)\|^2}{2} \geq \frac{L_0}{20 \max\{1, L_1^2\}}.$$

When $\epsilon \leq \|\nabla f(x_k)\| \leq L_0/L_1$, we have

$$\frac{h_k\|\nabla f(x_k)\|^2}{2} \geq \frac{\|\nabla f(x_k)\|^2}{20L_0} \geq \frac{\epsilon^2}{20L_0}.$$

Therefore,

$$f(x_{k+1}) \leq f(x_k) - \min\left\{\frac{L_0}{20\max\{1, L_1^2\}}, \frac{\epsilon^2}{20L_0}\right\}.$$

Assume that $\epsilon \leq \|\nabla f(x_k)\|$ for $k \leq T$ iterations. By doing a telescopic sum, we get

$$\sum_{k=0}^{T-1} f(x_{k+1}) - f(x_k) \leq -T\min\left\{\frac{L_0}{20\max\{1, L_1^2\}}, \frac{\epsilon^2}{20L_0}\right\}.$$

Rearranging we get

$$T \leq \frac{20L_0(f(x_0) - f^*)}{\epsilon^2} + \frac{20\max\{1, L_1^2\}(f(x_0) - f^*)}{L_0}.$$

## D   PROOF OF THEOREM 4

We will prove a lower bound for the iteration complexity of GD with fixed step size. The high level idea is that if GD converges for all functions satisfying the assumptions, then the step size needs to be small. However, this small step size will lead to very slow convergence for another function.

Recall that the fixed step size GD algorithm is parameterized by the scaler: step size $h$. **First, we show that when $h > \frac{2\log(M)+2}{ML_1}$,**

$$\sup_{\substack{x_0 \in \mathbb{R}^d, \\ f \in \mathcal{F}}} T_\epsilon(A_h[f, x_0], f) = \infty$$

We start with a function that grows exponentially. Let $L_1 > 1, M > 1$ be fixed constants. Pick the initial point $x_0 = (\log(M) + 1)/L_1$. Let the objective be defined as follows,

$$f(x) = \begin{cases} \frac{e^{-L_1 x}}{L_1 e}, & \text{for } x < -\frac{1}{L_1}, \\ \frac{L_1 x^2}{2} + \frac{1}{2L_1}, & \text{for } x \in [-\frac{1}{L_1}, \frac{1}{L_1}], \\ \frac{e^{L_1 x}}{L_1 e}, & \text{for } x > \frac{1}{L_1}. \end{cases}$$

We notice that the function satisfies the assumptions with constants

$$L_0 = 1, \quad L_1 > 1, \quad M > 1. \tag{11}$$

When $h > 2x_0/M$, we would have $|x_1| > |x_0|$. By symmetry of the function and the super-linear growth of the gradient norm, we know that the iterates will diverge. Hence, in order for gradient descent with a fixed step size $h$ to converge, $h$ must be small enough. Formally,

$$h \leq \frac{2x_0}{M} = \frac{2\log(M) + 2}{ML_1}.$$

**Second, we show that when $h \leq \frac{2\log(M)+2}{ML_1}$,**

$$\sup_{\substack{x_0 \in \mathbb{R}^d, \\ f \in \mathcal{F}}} T_\epsilon(A_h[f, x_0], f) \geq \Delta L_1 M/(4\epsilon^2(\log M + 1))$$

Now, let's look at a different objective that grows slowly.

$$f(x) = \begin{cases} -2\epsilon(x+1) + \frac{5\epsilon}{4}, & \text{for } x < -1, \\ \frac{\epsilon}{4}(6x^2 - x^4), & \text{for } x \in [-1, 1], \\ 2\epsilon(x-1) + \frac{5\epsilon}{4}, & \text{for } x > 1. \end{cases}$$

This function is also second order differentiable and satisfies the assumptions with constants in (11). If we set $x_0 = 1 + \Delta/\epsilon$ for some constant $\Delta > 0$, we know that $f(x_0) - f^* = 2\Delta + 5\epsilon/4$. With the step size choice $h \le (2 \log M + 2)/(M L_1)$, we know that in each step, $x_{k+1} \ge x_k - (4\epsilon(\log M + 1))/(L_1 M)$. Therefore, for $k \le \Delta L_1 M/(4\epsilon^2(\log M + 1))$,

$$\|\nabla f(x_k)\| = 2\epsilon.$$

After combining these two points, we proved the theorem by definition (9).

## E    PROOF OF THEOREM 6

We start by parametrizing the function value along the update,

$$f(\gamma(t)) := f(x_k - th\nabla f(x_k)), t \in [0, 1].$$

Note that with this parametrization, we have $\gamma(0) = x_k, \gamma(1) = x_{k+1}$. Now we would like to argue that if $f(x_k) \le f(x_0)$, then $\|\nabla f(x(t))\| \le M, \forall t \le 1$. Assume by contradiction that this is not true. Then there exists $\epsilon > 0, t \in [0, 1]$ such that $\|\nabla f(x(t))\| \ge M + \epsilon$. Since $\epsilon$ can be made arbitrarily small below a threshold, we assume $\epsilon < M$. Denote

$$t^* = \inf\{t \mid \|\nabla f(x(t))\| \ge M + \epsilon\}.$$

The value $t^*$ exists by continuity of $\|\nabla f(x(t))\|$ as a function of $t$. Then we know by Assumption 4 that $f(x(t^*)) > f(x_k)$. However, by Taylor expansion, we know that

$$f(x(t^*)) \le f(x_k) - th\|\nabla f(x_k)\|^2 + (th)^2\|\nabla f(x_k)\|^2 \int_0^t \|\nabla^{(2)} f(x(\tau))\| d\tau$$

$$\le f(x_k) - th\|\nabla f(x_k)\|^2 + (th)^2\|\nabla f(x_k)\|^2 (L_1(M + \epsilon) + L_0)$$

$$\le f(x_k).$$

The last inequality follows by $h = 1/(2(ML_1 + L_0))$. Hence we get a contradiction and conclude that for all $t \le 1$, $\|\nabla f(x(t))\| \le M$. Therefore, following the above inequality and Assumption 3, we get

$$f(x_{k+1}) \le f(x_k) - h\|\nabla f(x_k)\|^2 + h^2 \frac{L_1 M + L_0}{2}\|\nabla f(x_k)\|^2$$

$$\le f(x_k) - \frac{\epsilon^2}{4(ML_1 + L_0)}.$$

The conclusion follows by the same argument as in Theorem 3 via a telescopic sum over $k$.

## F    PROOF OF THEOREM 7

Recall that we set the following parameters

$$h_k = \min\{\frac{1}{16\eta L_1(\|g_k\| + \tau)}, \eta\}$$

$$\eta = \min\{\frac{1}{20L_0}, \frac{1}{128L_1\tau}, \frac{1}{\sqrt{T}}\} \tag{12}$$

Similar to proof of Theorem 3, we have

$$\mathbb{E}[f(x_{k+1})|] \le f(x_k) - \mathbb{E}[h_k\langle g_k, \nabla f(x_k)\rangle] + \frac{5L_0 + 4L_1\|\nabla f(x_k)\|}{2}\mathbb{E}[h_k^2\|g_k\|^2] \tag{13}$$

$$\le f(x_k) - \mathbb{E}[h_k\langle g_k, \nabla f(x_k)\rangle] + \frac{5L_0 + 4L_1\|\nabla f(x_k)\|}{2}\mathbb{E}[h_k^2(\|\nabla f(x_k)\|^2$$

$$+ \|g_k - \nabla f(x_k)\|^2 + 2\langle \nabla f(x_k), g_k - \nabla f(x_k)\rangle)]$$

$$\le f(x_k) + \mathbb{E}[-h_k + \frac{5L_0 + 4L_1\|\nabla f(x_k)\|}{2}h_k^2]\|\nabla f(x_k)\|^2$$

$$+ \mathbb{E}[h_k(-1 + (5L_0 + 4L_1\|\nabla f(x_k)\|)h_k)\langle \nabla f(x_k), g_k - \nabla f(x_k)\rangle]$$

$$+ \frac{5L_0 + 4L_1\|\nabla f(x_k)\|}{2}\mathbb{E}[h_k^2(\|g_k - \nabla f(x_k)\|^2)]$$

First we show $(5L_0 + 4L_1\|\nabla f(x_k)\|)h_k \le \frac{1}{2}$. This follows by $5L_0 h_k \le \frac{1}{4}, h_k 4L_1\|\nabla f(x_k)\| \le h_k 4L_1(\|g_k\| + \tau) \le \frac{1}{4}$. Substitute in (15) and we get

$$
\mathbb{E}[f(x_{k+1})|] \le f(x_k) + \mathbb{E}[-\frac{3h_k}{4}]\|\nabla f(x_k)\|^2 \tag{14}
$$
$$
+ \underbrace{\mathbb{E}[-h_k\langle\nabla f(x_k), g_k - \nabla f(x_k)\rangle]}_{T_1}
$$
$$
+ \underbrace{\mathbb{E}[(5L_0 + 4L_1\|\nabla f(x_k)\|)h_k^2\langle\nabla f(x_k), g_k - \nabla f(x_k)\rangle]}_{T_2}
$$
$$
+ \underbrace{\frac{5L_0 + 4L_1\|\nabla f(x_k)\|}{2}\mathbb{E}[h_k^2(\|g_k - \nabla f(x_k)\|^2)]}_{T_3}
$$

Then we bound $T_1, T_2, T_3$ in Lemma 11,12,13 and get

$$
\mathbb{E}[f(x_{k+1})|] \le f(x_k) + \mathbb{E}[-\frac{h_k}{4}]\|\nabla f(x_k)\|^2 \tag{15}
$$
$$
+ (5L_0 + 2L_1\tau)\eta^2\tau^2 + 9\eta^2\tau L_0^2/L_1
$$

Rearrange and do a telescopic sum, we get

$$
\mathbb{E}[\sum_{k\le T}\frac{h_k}{4}\|\nabla f(x_k)\|^2] \le f(x_0) - f^* + \eta^2 T((5L_0 + 2L_1\tau)\tau^2 + 9\tau L_0^2/L_1)
$$
$$
\le f(x_0) - f^* + ((5L_0 + 2L_1\tau)\tau^2 + 9\tau L_0^2/L_1)
$$

Furthermore, we know

$$
h_k\|\nabla f_k\|^2 = \min\{\eta, \frac{1}{16L_1(\|\nabla f_k\| + \tau)}\}\|\nabla f_k\|^2
$$
$$
\ge \min\{\eta, \frac{1}{32L_1\|\nabla f_k\|}, \frac{1}{32L_1\tau}\}\|\nabla f_k\|^2
$$
$$
\ge \min\{\eta, \frac{1}{32L_1\|\nabla f_k\|}\}\|\nabla f_k\|^2
$$

Hence along with $\eta \le T^{-1/2}$, we get

$$
\mathbb{E}[\sum_{k\le T}\min\{\eta\|\nabla f_k\|^2, \frac{\|\nabla f_k\|}{32L_1}\}] \le f(x_0) - f^* + ((5L_0 + 2L_1\tau)\tau^2 + 9\tau L_0^2/L_1)
$$

Let $\mathcal{U} = \{k|\eta\|\nabla f_k\|^2 \le \frac{\|\nabla f_k\|}{16L_1}\}$, we know that

$$
\mathbb{E}[\sum_{k\in\mathcal{U}}\eta\|\nabla f_k\|^2] \le f(x_0) - f^* + ((5L_0 + 2L_1\tau)\tau^2 + 9\tau L_0^2/L_1),
$$

and

$$
\mathbb{E}[\sum_{k\in\mathcal{U}^c}\frac{\|\nabla f_k\|}{32L_1}] \le f(x_0) - f^* + ((5L_0 + 2L_1\tau)\tau^2 + 9\tau L_0^2/L_1).
$$

Therefore,

$$
\mathbb{E}[\min_k\|\nabla f(x_k)\|] \le \mathbb{E}[\min\{\frac{1}{|\mathcal{U}|}\sum_{k\in\mathcal{U}}\|\nabla f_k\||, \frac{1}{|\mathcal{U}^c|}\sum_{k\in\mathcal{U}^c}\|\nabla f_k\|\}]
$$
$$
\le \mathbb{E}[\min\{\sqrt{\frac{1}{|\mathcal{U}|}\sum_{k\in\mathcal{U}}\|\nabla f_k\|^2}, \frac{1}{|\mathcal{U}^c|}\sum_{k\in\mathcal{U}^c}\|\nabla f_k\|\}]
$$
$$
\le \max\{\sqrt{f(x_0) - f^* + ((5L_0 + 2L_1\tau)\tau^2 + 9\tau L_0^2/L_1)}\frac{\sqrt{T} + 20L_0 + 128L_1\tau}{T},
$$
$$
(f(x_0) - f^* + (5L_0 + 2L_1\tau)\tau^2 + 9\tau L_0^2/L_1)\frac{64L_1}{T}\}.
$$

The last inequality follow by the fact that either $|\mathcal{U}| \geq T/2$ or $|\mathcal{U}^c| \geq T/2$. This implies that $\mathbb{E}[\min_{k \leq T} \|\nabla f(x_k)\|] \leq 2\epsilon$ when

$$T \geq \Delta \max\{\frac{128L_1}{\epsilon}, \frac{4\Delta}{\epsilon^4}, \frac{80L_0 + 512L_1\tau}{\epsilon^2}\},$$

where $\Delta = (f(x_0) - f^* + (5L_0 + 2L_1\tau)\tau^2 + 9\tau L_0^2/L_1)$. By Markov inequality,

$$\mathbb{P}\{\min_{k \leq T} \|\nabla f(x_k)\| \leq \epsilon\} \geq \frac{1}{2}.$$

The theorem follows by the definition in (9).

### F.1 TECHNICAL LEMMAS

**Lemma 11.**
$$\mathbb{E}[-h_k\langle \nabla f(x_k), g_k - \nabla f(x_k)\rangle] \leq \frac{1}{4}\mathbb{E}[h_k]\|\nabla f(x_k)\|^2.$$

*Proof.* By unbiasedness of $g_k$ and the fact that $\eta$ is a constant, we have

$$\mathbb{E}[-h_k\langle \nabla f(x_k), g_k - \nabla f(x_k)\rangle] = \mathbb{E}[(\eta - h_k)\langle \nabla f(x_k), g_k - \nabla f(x_k)\rangle]$$
$$= \mathbb{E}[(\eta - h_k)\langle \nabla f(x_k), g_k - \nabla f(x_k)\rangle \mathbb{1}_{\{\|g_k\| \geq \frac{1}{16L_1\eta} - \tau\}}]$$
$$\leq \eta\|\nabla f(x_k)\|\mathbb{E}[\|g_k - \nabla f(x_k)\|\mathbb{1}_{\{\|g_k\| \geq \frac{1}{16L_1\eta} - \tau\}}]$$
$$\leq \eta\|\nabla f(x_k)\|^2 32L_1\mathbb{E}[h_k]\tau$$

The second last inequality follows by $h_k \leq \eta$ and Cauchy-Schwartz inequality. The last inequality follows by

$$\|\nabla f(x_k)\| \geq \|g_k\| - \tau = \frac{1}{16L_1 h_k} - 2\tau \geq \frac{1}{16L_1 h_k} - \frac{1}{32L_1\eta} \geq \frac{1}{32L_1 h_k}. \tag{16}$$

The equality above holds because $h_k = \frac{1}{16\eta L_1(\|g_k\| + \tau)}$. The lemma follows by $32\eta L_1\tau \leq 1/4$. $\square$

**Lemma 12.**
$$\mathbb{E}[(5L_0 + 4L_1\|\nabla f(x_k)\|)h_k^2\langle \nabla f(x_k), g_k - \nabla f(x_k)\rangle] \leq 9\eta^2\tau L_0^2/L_1 + \frac{1}{8}\mathbb{E}[h_k]\|\nabla f(x_k)\|^2.$$

*Proof.* When $\|\nabla f(x_k)\| \geq L_0/L_1$,
$$\mathbb{E}[(5L_0 + 4L_1\|\nabla f(x_k)\|)h_k^2\langle \nabla f(x_k), g_k - \nabla f(x_k)\rangle] \leq 9L_1\|\nabla f(x_k)\|\mathbb{E}[h_k^2]\|\nabla f(x_k)\|\tau$$
$$\leq \frac{1}{8}\mathbb{E}[h_k]\|\nabla f(x_k)\|^2$$

The last inequality follows by (12).

When $\|\nabla f(x_k)\| \leq L_0/L_1$,
$$\mathbb{E}[(5L_0 + 4L_1\|\nabla f(x_k)\|)h_k^2\langle \nabla f(x_k), g_k - \nabla f(x_k)\rangle] \leq 9\eta^2\tau L_0^2/L_1$$

$\square$

**Lemma 13.**
$$\frac{5L_0 + 4L_1\|\nabla f(x_k)\|}{2}\mathbb{E}[h_k^2(\|g_k - \nabla f(x_k)\|^2)] \leq (5L_0 + 2L_1\tau)\eta^2\tau^2 + \frac{1}{8}\|\nabla f(x_k)\|^2\mathbb{E}[h_k].$$

*Proof.* When $\|\nabla f(x_k)\| \geq L_0/L_1 + \tau$, we get
$$\frac{5L_0 + 4L_1\|\nabla f(x_k)\|}{2}\mathbb{E}[h_k^2(\|g_k - \nabla f(x_k)\|^2)] \leq 5L_1\|\nabla f(x_k)\|^2\mathbb{E}[h_k]\eta\tau \leq \frac{1}{8}\|\nabla f(x_k)\|^2\mathbb{E}[h_k].$$
The first inequality follows by $h_k \leq \eta$ and $\|g_k - \nabla f(x_k)\| \leq \tau \leq \|\nabla f(x_k)\|$. The last inequality follows by (12).

When $\|\nabla f(x_k)\| \leq L_0/L_1 + \tau$, we get
$$\frac{5L_0 + 4L_1\|\nabla f(x_k)\|}{2}\mathbb{E}[h_k^2(\|g_k - \nabla f(x_k)\|^2)] \leq (5L_0 + 2L_1\tau)\eta^2\tau^2.$$

$\square$

## G  PROOF OF THEOREM 8

Similar to proof of Theorem 3, we have

$$\mathbb{E}[f(x_{k+1})|] \leq f(x_k) - \mathbb{E}[h_k\langle g_k, \nabla f(x_k)\rangle] + \frac{5L_0 + 4L_1\|\nabla f(x_k)\|}{2}\mathbb{E}[h_k^2\|g_k\|^2]$$

$$\leq f(x_k) - \frac{1}{\sqrt{T}}\|\nabla f(x_k)\|^2 + \frac{5L_0 + 4L_1M(M+\tau)^2}{2T}$$

Sum across $k \in \{0, ..., T-1\}$ and take expectations, then we can get

$$0 \leq f(x_0) - \mathbb{E}[f(x_T)] - \frac{1}{\sqrt{T}}\sum_{k=1}^{T}\mathbb{E}\Big[\|\nabla f(x_k)\|^2\Big] + \frac{5L_0 + 4L_1M(M+\tau)^2}{2}$$

Rearrange and we get

$$\frac{1}{T}\sum_{k=1}^{T}\mathbb{E}\Big[\|\nabla f(x_k)\|^2\Big] \leq \frac{1}{\sqrt{T}}\bigg(f(x_0) - f^* + \frac{5L_0 + 4L_1M(M+\tau)^2}{2}\bigg)$$

By Jensen's inequality,

$$\frac{1}{T}\sum_{k=1}^{T}\mathbb{E}[\|\nabla f(x_k)\|] \leq \sqrt{\frac{1}{\sqrt{T}}\bigg(f(x_0) - f^* + \frac{5L_0 + 4L_1M(M+\tau)^2}{2}\bigg)}$$

By Markov inequality,

$$\mathbb{P}\bigg\{\frac{1}{T}\sum_{k=1}^{T}\Big[\|\nabla f(x_k)\|^2\Big] > \frac{2}{\sqrt{T}}\bigg(f(x_0) - f^* + \frac{5L_0 + 4L_1M(M+\tau)^2}{2}\bigg)\bigg\} \leq 0.5$$

The theorem follows by the definition in (9) and Jensen's inequality.

## H  EXPERIMENT DETAILS

In this section, we first briefly overview the tasks and models used in our experiment. Then we explain how we estimate smoothness of the function. Lastly, we describe some details for generating the plots in Figure 2 and Figure 3.

### H.1  LANGUAGE MODELLING

Clipped gradient descent was introduced in (vanilla) recurrent neural network (RNN) language model (LM) (Mikolov et al., 2010) training to alleviate the *exploding gradient* problem, and has been used in more sophisticated RNN models (Hochreiter and Schmidhuber, 1997) or seq2seq models for language modelling or other NLP applications (Sutskever et al., 2014; Cho et al., 2014). In this work we experiment with LSTM LM (Sundermeyer et al., 2012), which has been an important building block for many popular NLP models (Young et al., 2017).

The task of language modelling is to model the probability of the next word $w_{t+1}$ based on word history (or context). Given a document of length $T$ (words) as training data, the training objective is to minimize negative log-likelihood of the data $\frac{-1}{T}\Sigma_{t=1}^{T}\log P(w_t|w_1...w_{t-1})$.

We run LM experiments on the Penn Treebank (PTB) (Mikolov et al., 2010) dataset, which has been a popular benchmark for language modelling. It has a vocabulary of size 10k, and 887k/70k/78k words for training/validation/testing.

To train the LSTM LM, we follow the training recipe from [3] (Merity et al., 2018). The model is a 3-layer LSTM LM with hidden size of 1150 and embedding size of 400. Dropout (Srivastava et al., 2014) of rate 0.4 and DropConnect (Wan et al., 2013) of rate 0.5 is applied. For optimization, clipped SGD with clip value of 0.25 and a learning rate of 30 is used, and the model is trained for 500 epochs. After training, the model reaches a text-set perplexity of 56.5, which is very close to the current state-of-art result (Dai et al., 2019) on the PTB dataset.

---

[3]https://github.com/salesforce/awd-lstm-lm

## H.2 IMAGE CLASSIFICATION

As a comparison, we run the same set of experiments on image classification tasks. We train the ResNet20 (He et al., 2016) model on Cifar10 (Krizhevsky and Hinton, 2009) classification dataset. The dataset contains 50k training images and 10k testing images in 10 classes.

Unless explicitly state, we use the standard hyper-parameters based on the Github repository[4]. Our baseline algorithm runs SGD momentum with learning rate 0.1, momentum 0.9 for 200 epochs. We choose weight decay to be $5e-4$. The learning rate is reduced by 10 at epoch 100 and 150. Up to our knowledge, this baseline achieves the best known test accuracy (95.0%) for Resnet20 on Cifar10. The baseline already beats some recently proposed algorithms which claim to improve upon SGD momentum.

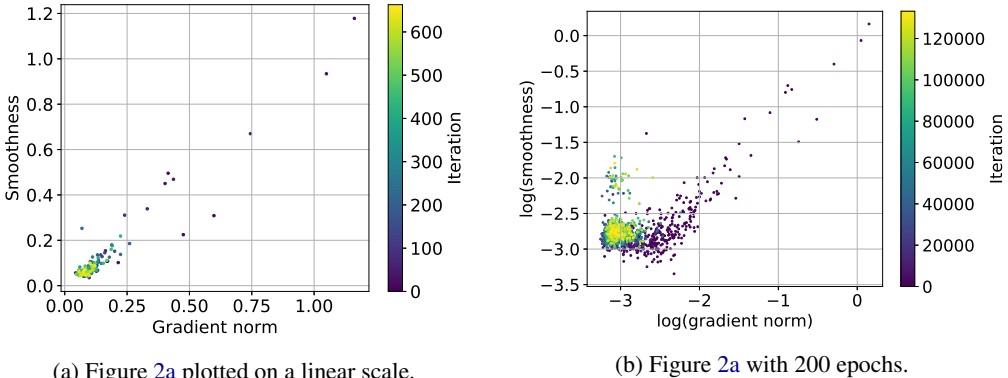

(a) Figure 2a plotted on a linear scale.

(b) Figure 2a with 200 epochs.

Figure 5: Auxiliary plots for Figure 2a. The left subfigure shows the values scattered on linear scale. The right subfigure shows more data points from 200 epochs.

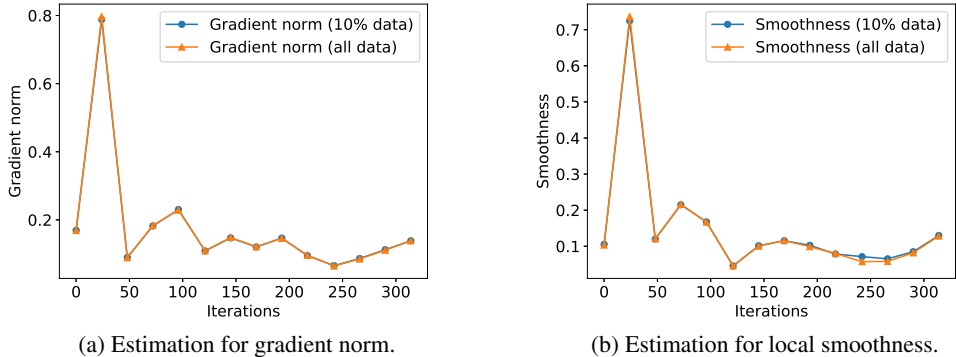

(a) Estimation for gradient norm.

(b) Estimation for local smoothness.

Figure 6: Estimated gradient norm and smoothness using 10% data versus all data. The values are computed from checkpoints of the LSTM LM model in the first epoch. This shows that statistics evaluated from 10% of the entire dataset provides accurate estimation.

## H.3 ESTIMATING SMOOTHNESS

Our smoothness estimator follows a similar implementation as in (Santurkar et al., 2018). More precisely, given a sequence of iterates generated by training procedure $\{x_k\}_k$, we estimate the smoothness $\hat{L}(x_k)$ as follows. For some small value $\delta \in (0,1), d = x_{k+1} - x_k$,

$$\hat{L}(x_k) = \max_{\gamma \in \{\delta, 2\delta, \dots, 1\}} \frac{\|\nabla f(x + \gamma d) - \nabla f(x)\|}{\|\gamma d\|}. \tag{17}$$

---

[4]https://github.com/kuangliu/pytorch-cifar

This suggests that we only care about the variation of gradient along $x_{k+1} - x_k$. The motivation is based on the function upper bound (10), which shows that the deviation of the objective from its linear approximation is determined by the variation of gradient between $x_{k+1}$ and $x_k$.

### H.4 ADDITIONAL PLOTS

The plots in Figure 2a show log-scale scattered data for iterates in the first epoch. To supplement this result, we show in Figure 5a the linear scale plot of the same data as in Figure 2a. In Figure 5b, we run the same experiment as in Figure 2a for 200 epochs instead of 1 epoch and plot the gradient norm and estimated smoothness along the trajectory.

In Figure 2, we plot the correlation between gradient norm and smoothness in LSTM LM training. We take snapshots of the model every 5 iterations in the first epoch, and use 10% of training data to estimate gradient norm and smoothness. As shown in Figure 6, using 10% of the data provides a very accurate estimate of the smoothness computed from the entire data.

## I  A SYNTHETIC EXPERIMENT

In this section, we demonstrate the different behaviors of gradient descent versus clipped gradient descent by optimizing a simple polynomial $f(x) = x^4$. We initialize the point at $x_0 = 30$ and run both algorithms. Within the sublevel set $[-30, 30]$, the function satisfies

$$f''(x) \le 12 \times 30^2 = 1.08 \times 10^4$$
$$f''(x) \le 10 f'(x) + 0.1.$$

Therefore, we can either pick $L_1 = 0, L_0 = 1.08^4$ for gradient descent or $L_1 = 10. L_0 = 0.1$ for clipped GD. Since the theoretical analysis is not tight with respect to constants, we scan the step sizes to pick the best parameter for both algorithms. For gradient descent, we scan step size by halving the current steps. For clipped gradient descent, we fix threshold to be 0.01 and pick the step size in the same way. The convergence results are shown in Figure 7. We can conclude that clipped gradient descent converges much faster than vanilla gradient descent, as the theory suggested.

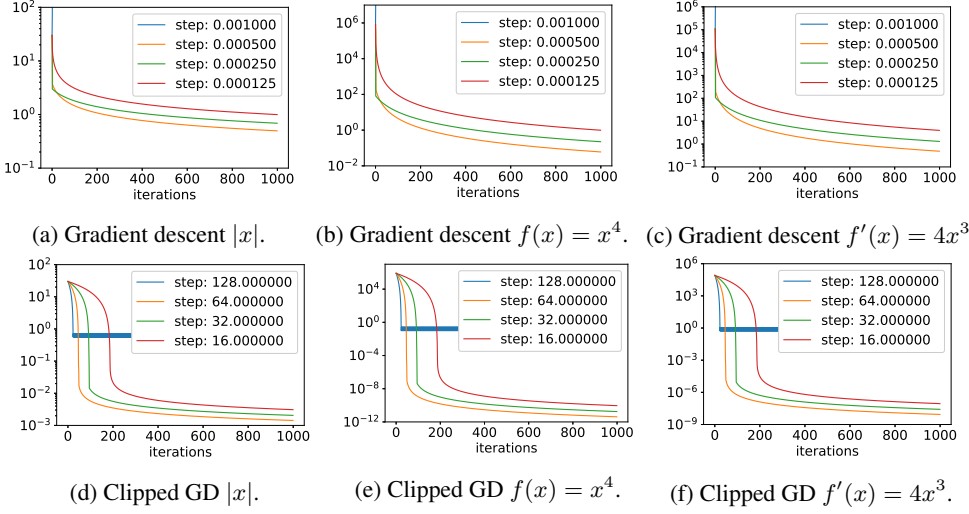

(a) Gradient descent $|x|$.  (b) Gradient descent $f(x) = x^4$.  (c) Gradient descent $f'(x) = 4x^3$.

(d) Clipped GD $|x|$.  (e) Clipped GD $f(x) = x^4$.  (f) Clipped GD $f'(x) = 4x^3$.

Figure 7: An synthetic experiment to optimize $f(x) = x^4$. (a) Gradient descent with different step size. (b) Clipped gradient descent with different step size and threshold $= 0.01$.

## J  A QUANTITATIVE COMPARISON OF THEOREMS AND EXPERIMENTS

To quantify how much the result align with the theorem, we assume that the leading term in the iteration complexity is the $\epsilon$ dependent term. For GD, the term scales as $\mathcal{O}(\frac{ML_1 + L_0}{\sqrt{T}})$, while for Clipped GD, the term scales as $\mathcal{O}(\frac{L_0}{\sqrt{T}})$.

First, we start with the synthetic experiment presented in I. From theory, we infer that the improvement of $\frac{f'(x_{GD})}{f'(x_{CGD})} \approx \frac{L_0}{ML_1+L_0} \approx 1e5$. In experiment, the best performing GD reaches $f'(x_T) = 0.36$, while for clipped GD, the value is $1.3e-8$, and the ratio is $\frac{f'(x_{GD})}{f'(x_{CGD})} \approx 1e7$. This suggests that in this very adversarial (against vanilla GD) synthetic experiment, the theorem is correct but conservative.

Next, we test how well the theory can align with practice in neural network training. To do so, we rerun the PTB experiment in 5 with a smaller architecture (2-Layer LSTM with 200 embedding dimension and 512 inner dimension). We choose hyperparameters based on Figure 4a. For clipped GD, we choose clipping threshold to be 0.25 and learning rate to be 10. For GD, we use a learning rate 2. One interesting observation is that, though GD makes steady progress in minimizing function value, its gradient norm is not decreasing.

To quantify the difference between theory and practice, we follow the procedure as in the synthetic experiment. First, we estimate $ML_1 + L_0 = 25$ for GD from Figure 8(c). Second, we estimate $L_1 = 10, L_0 = 5$ for clipped GD's trajectory from subplot (d). Then the theory predicts that the ratio between gradients should be roughly $25/5 = 5$. Empirically, we found the ratio to be $\approx 3$ by taking the average (as in Theorem 3 and Theorem 6). This doesn't exactly align but is of the same scale. From our view, the main reason for the difference could be the presence of noise in this experiment. As theorems suggested, noise impacts convergence rates but is absent in our rough estimates $\frac{ML_1+L_0}{L_0}$.

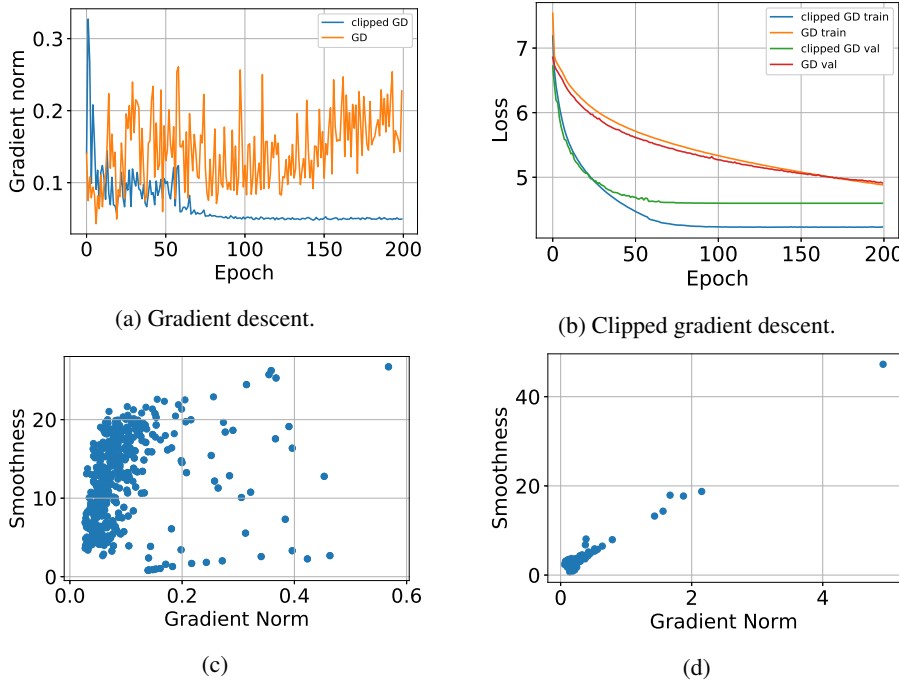

Figure 8: (a) Gradient norm. (b) Loss curves. (c)The scatter points of smoothness vs gradient norm for the model trained with gradient descent.(d)The scatter points of smoothness vs gradient norm for the model trained with clipped GD.

