# OpenReview forum: "Why Gradient Clipping Accelerates Training: A Theoretical Justification for Adaptivity"
_ICLR.cc/2020/Conference — Accept (Talk)_

### Official Review · AnonReviewer3 · 2019-10-16
**Official Blind Review #3**

**Rating:** 8

**Review:**

The authors provide the definition of a (L_0, L_1)-smooth function.
They motivate this class of function based on theoretical arguments and
empirical observations in the context of neural network training loss.
The authors go on to show that for (L_0, L_1)-smooth loss constant
step-size gradient descent can perform arbitrarily bad with poor initialization.
Specifically, the largest constant step size that can be used without divergence
depends on the upper bound of the gradient norm.
If the function is steep (high gradient norm) at some point during the
optimization process the steps need to become arbitrarily small
(especially relevant in nonconvex loss, where gradient norm does not always decrease).
The small constant step size slows down convergence everywhere else.
The described phenomenon is an argument for step sizes adaptivity in general
Clipping the gradient based on the known (L_0, L_1)-smoothness is a form of
adaptivity that makes steps small enough to avoid divergence only where the
function is steep and can thus be more efficient.

Overall, the authors give a theoretical justification for the use of clipped
gradient descent in the context of training neural networks.
The analysis of clipped gradient descent is performed rigorously (proofs are
provided in the appendix) and extended to the stochastic gradient descent setting.
The motivation for using (L_0, L_1)-smoothness and gradient clipping is also
shown to be relevant in practical experiments with language and image
classification models.

I recommend to accept the paper.

The paper is a clearly stated contribution to optimization theory as motivated
by machine learning applications.
The authors give a clean theoretical analysis of a previously empirically
observed phenomenon and an algorithm that is already used in practice.

Notes:
- Page 1, first paragraph:
If f(x) = E_xi [f(x, xi)] then f(x) the expectation is generally not stochastic,
like you write before. It is clear what you mean but Maybe use a different symbol.

- Page 2, ...we observe the function smoothness has a strong correlation with gradient norm 2:
Perhaps write something like "(see Figure 2)" instead of just "2" here.

- Page 2, after definition of Lipschitz continuous:
Nitpick: Formal definition based on Hessian norm upper bound should also have, for all x \in \mathbb R^d

- General: Some equations that are not referenced seem to be numbered, some are not.


**Experience Assessment:**

I have read many papers in this area.

**Review Assessment: Checking Correctness Of Derivations And Theory:**

I assessed the sensibility of the derivations and theory.

**Review Assessment: Checking Correctness Of Experiments:**

I assessed the sensibility of the experiments.

**Review Assessment: Thoroughness In Paper Reading:**

I read the paper at least twice and used my best judgement in assessing the paper.

---

> ### Author Response · Authors · 2019-11-10
> **Thank you for the comments.**
>
> We thank the reviewer for the positive comments and helpful notes. The paper is updated as the reviewer suggested.

---

### Official Review · AnonReviewer2 · 2019-10-16
**Official Blind Review #2**

**Rating:** 8

**Review:**

In this paper, the authors relax the generally used Lipschitz smoothness condition in optimization, to a more general smoothness condition that may depend on norm of the gradient. The authors proved that, with this relaxed condition, under such cases, both GD and clipped GD can converge within O(1/\epsilon^2) time, but when their exists x where the gradient norm can be very large, the clipped GD will converge provably faster than GD with a proved GD convergence rate lower bound. The authors also generalize their results to SGD. Experiments show that in deep neural network local gradient Lipschitz constant is scale with gradient norm, and use clipped gradient can accelerate convergence while keep good generalization performance as expected, which is well observed by other researchers.

Detailed comments:
1. The f^* In theorem 3 is not defined. I think it’s not the f^* in Assumption 1? Should be more clear. Also, in non-convex optimization, can the deterministic GD find the global optimum? If f^* is the stationary point the algorithm find, then the Assumption 1 is a little weird. The authors should better clarify this.
2. Feel the claim of the Assumption can be relaxed to only hold on \mathcal{S} is a little confusing. Which Assumption can be relaxed? It seems that in the proof of Lemma 9, we need the Assumption 3 holds for the set \|x^+-x\| \leq min{1/L_1, 1}. I think maybe the other Assumption will also be needed globally, unless the authors can prove that the optimization is only on a compact set in S.
3. Maybe a discussion connected the existing Lipschitz constant based results to the new results in this paper can make the readers more aware of the contribution in this paper. For example, the Lipschitz constant of neural network can be of order O(L_1 M) due to the definition, so if L_1 M is large, for clipped GD we can still have expected decreasing of O(1/L_0) each turn while vanilla GD with the best O(1/L) step size can still only have O(1/L) where L = L_0 + L_1 M due to the traditional results based on Lipschitz smoothness.

I don’t totally go through the whole proof, but I think the results are reasonable and the most of the techniques are standard to the whole community. So there may be no fatal error that violates the conclusion. I find the whole paper interesting and match the intuition from the practice. However, I think it can be polished to make the whole idea more clear and more easy to accept by the potential theorists and practitioner audience.

**Experience Assessment:**

I have read many papers in this area.

**Review Assessment: Checking Correctness Of Derivations And Theory:**

I assessed the sensibility of the derivations and theory.

**Review Assessment: Checking Correctness Of Experiments:**

I did not assess the experiments.

**Review Assessment: Thoroughness In Paper Reading:**

I read the paper thoroughly.

---

> ### Author Response · Authors · 2019-11-10
> **Clarification on theorems.**
>
> We would like to thank the reviewer for the positive feedback and comments. As the reviewer pointed out, our goal is to bridge the gap of theoretical result and practice observations. Hence, the reviewer's suggestion on an additional discussion is much appreciated.
>
> (1) f^* is the global minimum of the nonconvex function. For example, in most neural network training, f^*  = 0 as the network usually is allowed to overfit. However, though f^* is the global minimum, we do not provide bound for convergence of GD to global minimum. Instead, the bound is on gradient norm, which suggests that GD converges to stationary points. In other words, we provide a guarantee for convergence to stationary points instead of global min, though the number of iterations depends on the value of global min. This dependence is required as shown by [Y. Carmon, J. C. Duchi, O. Hinder, and A. Sidford. Lower bounds for finding stationary points].
>
> (2) We apologize for the confusion. In the deterministic case, all the assumptions only need to hold in \mathcal{S}. In the proof, we show that the function value is nonincreasing in each iteration. Therefore, even though we are solving an unconstrained problem, the iterates never escape the set \mathcal{S}.
>
> (3) We appreciate the reviewer’s suggestion and will edit accordingly when we have an extra page in the camera-ready version.

---

> > ### Comment · AnonReviewer2 · 2019-11-14
> > **Thanks for the clarification.**
> >
> > I hope the authors can also make these clarifications in the paper so that the potential audience need not to suffer from the questions I first have. Thanks!

---

### Official Review · AnonReviewer1 · 2019-10-25
**Official Blind Review #1**

**Rating:** 8

**Review:**

This paper applies new assumption on smoothness that assume the norm of Hessian is bounded by a scalar plus norm of gradient. The traditional smoothness is only bounded with a scalar, the proposed assumption is more relaxed because now the norm of Hessian can grow with the norm of gradient. Under this assumption, the authors show clipped gradient converges faster than gradient for general nonconvex problems. The authors provide insights on why the proposed assumption is good for describing neural networks, and empirically verify the assumption with ResNet on CIFAR and LSTM on PTB.

I like the paper in general. It is well written and easy to follow. The contributions are clearly described and the techniques seem to be solid.

I want a little bit more discussion to help me better understand the paper.

1) Intuitively, try best in plain English, why does clipped gradient convergence does not have dependency on L_1 M? Could the authors provide more discussion on the learning rate (hyperparameters) used for clipped gradient and gradient?

2) The convergence rates under the proposed assumption are slower than traditional smoothness assumption. Could you verify the slow convergence rate under proposed assumption aligns better with practical training? Could you provide a toy example, for example x^3, to show the advantage of the proposed assumption and the convergence under the assumption? What is the gap between clipped gradient and gradient, for experiments in figure 4, and possible toy problem? Could the authors elaborate the difference of clipped gradient and gradient with the details of theory, instead of simply claiming clipped gradient is faster?

Typo, page 5, (0, L) -> (L, 0)


============ after rebuttal ===================
I am happy to see the paper in the conference. The paper is intuitive and well written. My remaining comments are still on the slower convergence rate comparing to previous assumption. Even before, researchers suggest overparameterized network can converge faster than expected. This paper suggests a generally slower convergence rate if the proposed assumption does fit practice better.


**Experience Assessment:**

I have published one or two papers in this area.

**Review Assessment: Checking Correctness Of Derivations And Theory:**

I assessed the sensibility of the derivations and theory.

**Review Assessment: Checking Correctness Of Experiments:**

I assessed the sensibility of the experiments.

**Review Assessment: Thoroughness In Paper Reading:**

I read the paper at least twice and used my best judgement in assessing the paper.

---

> ### Author Response · Authors · 2019-11-10
> **Intuitions; A synthetic experiment added in Appendix I; Some clarifications needed**
>
> We would like to thank the reviewer for the feedback and comments. We address the reviewer’s questions as follows:
>
> (1) Intuition for the faster convergence of clipped GD: On a high level, gradient descent with a fixed step size has an iteration dependency on (L1M/\epsilon) because the algorithm needs to use a very small step size in the steep regions, which will slow down convergence in the flat regions, which determines the $\epsilon$ dependent term. This problem is circumvented by clipping because adaptivity allows the gradient descent to automatically use a small step size in steep regions with large gradients. When the iterates enter a flat region, larger learning rates are adopted if the gradient is small. As a result, the convergence rates in the flat region and in the steep region are decoupled as a sum instead of a multiplication.
>
> (2) Hyperparameters: Gradient clipping indeed has one more parameter (clipping threshold) to tune than GD. This parameter determines when the clipping effect should kick in. In fact, within a compact set, for each L1, there is a corresponding L0 to make the assumption hold. In practice, we usually pick one or two values for clipping threshold, and then scan the step size.
>
> (3) Slow convergence due to relaxed assumption: We thank the reviewer for the suggestion. A synthetic experiment is now included in Appendix I.
>
> Since we observe empirically that the local smoothness of a neural network is positively correlated with gradient norm, therefore, our theorem 3 and 6 jusitify the gap by proving that clipped GD converges faster. We do not fully understand what the reviewer mean by the “the gap between clipped gradient and gradient ”. Is the reviewer asking what is the proportion of gradients that are clipped? We are willing to supplement with experiments if the reviewer could elaborate more on this question.

---

> > ### Comment · AnonReviewer1 · 2019-11-13
> > **clarification on “the gap between clipped gradient and gradient ”**
> >
> > My question is whether the empirical difference of clipped gradient and gradient aligns with the theoretical difference presented in a probably more quantitative way. There are several curves in figure 4, I am trying to get more info.  Sorry if it makes any confusion.

---

> > > ### Author Response · Authors · 2019-11-15
> > > **New experiment and analysis updated in Appendix J**
> > >
> > > Thank you very much for the clarification!
> > >
> > > We did an analysis to quantify the gap between the theory and the experiments for both synthetic data and PTB language modeling. The details are in Appendix J of the updated paper. We found that the theory predictions,  though not precise, are on the same scale as the experiments.
> > >
> > > We thank the reviewer for raising this very interesting question.

---

### Author Response · Authors · 2019-11-10
**Paper updated with an additional synthetic experiments.**

We thank the reviewers for the positive feedbacks and helpful comments. We believe that aligning the assumptions of theoretical analysis to explain/predict practice is an exciting direction for studying optimization.

Based on the comments, we included an additional synthetic experiment for optimizing polynomial in Appendix I. We also corrected some typos according to Reviewer 3's feedback.

---

> ### Author Response · Authors · 2019-11-15
> **Updated again to quantify the gap between theorem and experiment results (Appendix J)**
>
> We updated the paper (see Appendix J) based on reviewer 1's comments. In particular, we show that the gap predicted by theory is on the same scale as the gap we observe in experiments (including PTB the Language modeling experiment). We thank the reviewer 1 for this interesting suggestion and enjoyed doing the experiments.
>
> We also added some clarifications based on Reviewer 2's helpful suggestions.

---

### Public Comment · ~Kyunghun_Nam1 · 2023-11-17
**Several questions for your paper**

Dear,

Great read of your wonderful paper.
I have a question that I would like to ask you. If you answer it, it will deepen my understanding.

My question is about Remark 5.
First of all, let me ask you a trivial question.
In the lower bound term, is $\log(M) +L_0$ in the denominator together? Or is only $\log(M)$ in the denominator?
I'm guessing it's the former, but the way it's written, it's the latter, so I'm asking.

Here's an important question.
There is an argument that there is a lower bound and an upper bound and that Clipped GD is faster than GD when $L_1M$ is large.
There is a statement that this happens when there is a poor initialization.

I don't really understand these two things, i.e., when $L_1M$ is large, the lower bound of the GD is larger than the upper bound of the Clipped GD.
Next, I wonder why $L_1M$ is large when has a poor initialization, and we also see $f(x_0) - f^{\star}$ term could be also large when the problem has a poor initialization.

These may be rudimentary questions, but I would appreciate a kind answer.

Best regards.

---

> ### Public Comment · ~Kyunghun_Nam1 · 2023-11-23
> **Additional question**
>
> In the page 13~14, the proof of theorem 3,
> I can't understand how the following inequality holds.
>
> When $\lVert \nabla f(x_k) \rVert \ge \frac{L_0}{L_1}$,
> $\frac{h_k \lVert \nabla f(x_k) \rVert^2}{2} \ge \frac{L_0}{20 \max\{1, L_1^2 \}}$.
>
> Since we don't know the lower bound of $h_k$, we can't derive the above lower bound.
>
> And I don't understand how the second case comes out either.
>
> When $\lVert \nabla f(x_k) \rVert \le \frac{L_0}{L_1}$,
> $\frac{h_k \lVert \nabla f(x_k) \rVert^2}{2} \ge \frac{\lVert \nabla f(x_k) \rVert^2}{20 L_0}$.
>
> In other words, I can't understand why $h_k \ge \frac{1}{10 L_0}$, and I would appreciate it if you could kindly explain if I missed it somewhere.
>
> Thank you.

---

> ### Public Comment · ~Kyunghun_Nam1 · 2023-11-28
> **Request**
>
> Still, I am waiting for your reply.

---

> ### Author Response · Authors · 2023-11-28
> **Response**
>
> Hi,
>
> We appreciate insightful questions but may not address all the technical ones if those questions were already answered in our manuscript.
>
> " First of all, let me ask you a trivial question..."
> The latter one.
>
> " I don't really understand these two things..."
> By poor initialization, we meant an $x$ where the local smoothness is large, and close to $L_1 M$
>
> Thanks

---

### Decision · Program_Chairs · 2019-12-19

**Decision:**

Accept (Talk)

**Comment:**

Gradient clipping is increasingly popular and it's nice to see a paper theoretically exploring its nice performance. All reviewers appreciated the work and the results.

Please make sure to incorporate all of their comments for the final version.